# Bridging the Gap: Two Decades of Childhood Vaccination Coverage and Equity in Cambodia and the Philippines (2000–2022)

**DOI:** 10.3390/vaccines13090907

**Published:** 2025-08-27

**Authors:** Yanqin Zhang, Xinyu Zhang, Qian Long

**Affiliations:** 1Global Health Research Center, Duke Kunshan University, Kunshan 215316, China; yanqin.zhang@dukekunshan.edu.cn (Y.Z.); xinyu.zhang@dukekunshan.edu.cn (X.Z.); 2Duke Global Health Institute, Duke University, Durham, NC 27708, USA

**Keywords:** child immunization, childhood vaccine, national immunization program, health equity, health policy

## Abstract

**Background/Objectives:** Equitable access to childhood vaccines remains a challenge in many low- and middle-income countries. This study assessed coverage of WHO-recommended childhood vaccines in Cambodia and the Philippines, focusing on urban–rural and wealth disparities, and examined maternal demographic and socioeconomic factors influencing vaccination coverage. **Methods:** Cross-sectional data from Demographic and Health Surveys from Cambodia (2000–2021/22) and the Philippines (2003–2022) were used. Descriptive analyses were performed to elucidate vaccination coverage trends (BCG, hepatitis B birth dose, DTP, OPV, PCV, and measles). Urban–rural and wealth-related disparities were assessed by calculating absolute differences and Slope Index of Inequality. Logistic regression was used to analyze the impact of maternal demographics and socioeconomic status on vaccination coverage. **Results:** Cambodia showed significant increases in BCG, DTP, and OPV coverage over the past two decades, whereas those coverage in the Philippines declined slightly since 2017. In 2022, 75.2% of Filipino children received the BCG and hepatitis B (birth dose) vaccines, and around two-thirds completed DTP, OPV, and PCV vaccinations on schedule, lower than the rates in Cambodia. Only half of the children completed measles vaccination in both countries. Urban–rural disparities declined over time in both countries, but wealth inequalities persisted and widened in the Philippines between 2017 and 2022. Women with higher education attainment, from a wealthy household and having fewer children, was associated with increased likelihood of completing childhood vaccinations in both countries. **Conclusions:** Persistent socioeconomic disparities in childhood vaccination in low- and middle-income countries highlight the need for targeted pro-poor and community-based strategies to ensure equitable access.

## 1. Introduction

Established in 1974 by the World Health Assembly, the Expanded Program on Immunization (EPI, now also referred to as the national immunization program, NIP) was originally designed to protect children from diseases such as diphtheria, tetanus, pertussis, measles, polio, tuberculosis, and smallpox—the only disease eradicated globally [1]. By 2024, the EPI is expected to include vaccines against 13 diseases across the life course worldwide [2]. According to recent World Health Organization (WHO) estimates, immunization has reduced infant deaths by 40% in the past five decades, with measles vaccination accounting for 60% of lives saved [1]. Vaccination remains one of the most cost-effective public health interventions. The Immunization Agenda 2030 (IA2030) highlights achieving 90% global coverage for key childhood vaccines —diphtheria–tetanus–pertussis (DTP), pneumococcal conjugate vaccine (PCV), and measles-containing vaccine (MCV)—by 2030, while reducing the number of zero-dose children, defined as children who have not received the first dose of the DTP vaccine by the end of their first year of life, by 50% compared to 2019 levels [3]. However, equitable access to childhood vaccination remains a critical challenge in many low- and middle-income countries (LMICs), where significant disparities persist, shaped by maternal demographic characteristics, household socioeconomic status, social norms, and cultural beliefs [4].

Since 2000, Gavi, The Vaccine Alliance, has been dedicated to improving equitable access to new and underutilized vaccines in low-resource settings by making them available and strengthening delivery systems [1,5]. Over the past two decades, it has focused on fostering political commitment, mobilizing domestic resources, and integrating vaccination into primary healthcare to ensure sustainable coverage [6]. However, middle-income countries (MICs), particularly those not eligible for Gavi support, face persistent challenges in achieving equitable immunization coverage. These challenges stem from systemic barriers such as policy decision-making, financing, demand generation, and access to affordable vaccines [7]. In 2020, Gavi introduced a tailored support initiative for middle-income countries (known as the Middle-Income Country Approach) to address systemic challenges, prevent coverage backsliding, and facilitate the sustainable introduction of vaccines [5].

Cambodia and the Philippines, both categorized as lower-middle-income countries, reported GDP per capita of USD 1875.1 and USD 3725.55 in 2023, respectively [5]. Cambodia has been a Gavi-eligible country since 2002 and began preparing to reduce financial reliance on external funding for its immunization program [5,8]. The Philippines has been eligible for support under Gavi’s MICs Approach, and its NIP is primarily funded by the government [5]. Both Cambodia and the Philippines included all WHO-recommended childhood vaccines except the rotavirus vaccine [5]. In both countries, public primary health facilities serve as the primary providers of immunization services, although private health facilities also play a contributing role [5]. Over the past decades, Cambodia and the Philippines have successfully expanded the coverage of their NIP. However, recent measles outbreaks in both countries underscore the need to revisit the immunization strategy for vaccine-preventable diseases, particularly for socially vulnerable groups [9,10]. This study examined vaccine coverage trends within the NIP of Cambodia and the Philippines over the past two decades, with a particular focus on urban–rural and socioeconomic disparities. Additionally, it analyzed maternal demographic and socioeconomic factors associated with key childhood vaccines coverage to inform strategies aimed at achieving equitable immunization coverage.

## 2. Materials and Methods

This study analyzed cross-sectional data from five waves of the Demographic and Health Surveys (DHSs) conducted in Cambodia (2000–2021/22) and the Philippines (2003–2022) [11]. Both countries used a two-stage stratified sampling design, selecting primary sampling units from provinces or enumeration areas defined in the General Population Census and systematically selecting households to ensure national and regional representativeness [12,13]. Data on immunization for children aged 0–23 months were drawn from women datasets, with records sourced from immunization cards or mothers’ recall. To avoid overrepresentation, only the most recent child was included when mothers had multiple children in this age range.

### 2.1. Outcome Measures

The outcome measures were vaccination coverage on schedule among children aged 0–23 months. According to national immunization schedules in both study countries, children within this age range could receive one dose of Bacillus Calmette-Guérin (BCG) and hepatitis B (at birth), three doses each of diphtheria–tetanus–pertussis (DTP), oral polio vaccine (OPV), and pneumococcal conjugate vaccine (PCV) (at ages of 6, 10, and 14 weeks), and two doses of measles vaccine (at ages of 9 months and 18 months in Cambodia and at ages of 9 months and 12 months in the Philippines) [12,13]. Both standalone DTP and pentavalent formulations were analyzed to study coverage of DTP.

Vaccination coverage was categorized as “yes” or “no” for single-dose vaccines, and as “no vaccination,” “incomplete,” or “completed” for multiple-dose vaccines. It was calculated by dividing the number of children who had never received, partially received, or fully received a specific vaccine by the total number of children who had reached the eligible age for that vaccine by 23 months of age. For multiple-dose vaccines, coverage was assessed among children who had reached the eligible age for the last dose.

### 2.2. Explanatory Variables

The explainable variables included maternal demographic and socioeconomic status including the following: place of residence (urban and rural), maternal age (≤19, 20–29, and 30+), education (no education, incomplete primary, primary, and secondary and higher), wealth index (poorest, poorer, middle, richer, and richest), employment (did not work and worked), and parity (1, 2, and 3+). The wealth index, constructed by the DHS team, is based in a composite measure of household living standards derived from ownership of selected assets, housing construction materials, and access to water and sanitation facilities [14].

### 2.3. Statistical Analysis

Data on BCG, DTP, and OPV were available in all five survey waves in both study countries. However, data on hepatitis B (birth dose), measles (second dose), and PCV (all three doses) were only available in the 2021–2022 (later refer to 2021) survey for Cambodia and the 2017 and 2022 surveys for the Philippines. Descriptive analyses were conducted to examine trends in vaccination coverage for each of the six childhood vaccines according to the schedule over time in both study countries, based on data availability. We further examined the absolute difference in vaccination coverage by urban–rural residence and wealth index groups. The Slope Index of Inequality (SII), which represents the absolute percentage point (pp) difference in coverage between the richest and poorest groups, was calculated using logistic regression analysis. Positive values indicate pro-rich inequality, while negative values reflect pro-poor patterns [15]. Multivariable logistic regression analyses were performed to investigate the associations between maternal characteristics and vaccination coverage for each vaccine, with a particular focus on identifying factors influencing coverage of DTP, PCV, and measles vaccines in the two study countries, in the context of the IA2030 targets [3]. All analyses were conducted using R version 4.3.3 [16].

## 3. Results

### 3.1. Participants

This study included 15,387 Cambodian and 14,562 Filipino women with children aged 0–23 months. Urban residency among Cambodian women rose from 13.8% in 2000 to 32.8% in 2021, while in the Philippines, it declined from 45.9% in 2003 to 35.3% in 2022 (Table 1).

In Cambodia, the proportion of women over 30 declined from 45.3% in 2000 to 38.7% in 2021, while in the Philippines, it increased from 39.1% in 2003 to 43.3% in 2022. By 2022, 63.0% of Filipino women had attained secondary or higher education, compared to 13.5% of Cambodian women in 2021. The poorest quintile constituted the largest socioeconomic group in both countries, representing 28.8% in Cambodia in 2021 and 35.8% in the Philippines in 2022. A total of 64.5% of Cambodian women were employed in 2021, while 39.3% of Filipino women were employed in 2022. The proportion of Cambodian women with three or more children dropped notably from 63.7% in 2000 to 30.9% in 2021, while in the Philippines, this figure remained stable at 43.8% in 2022 (Table 1).

### 3.2. Routine Childhood Vaccination Coverage on Schedule

Table 1 presents the number of children eligible for BCG, hepatitis B, DTP, OPV, PCV, and measles vaccines according to the schedule in both countries.

### 3.3. BCG and Hepatitis B (Single-Dose at Birth)

In Cambodia, BCG coverage rose from 59.4% in 2000 to 91.6% in 2021, with hepatitis B birth dose reaching 91.7% in the same year. In the Philippines, BCG coverage increased from 82.9% in 2003 to 91.0% in 2013, then declined to 83.5% by 2022. Hepatitis B birth dose coverage remained lower, improving slightly from 72.7% in 2017 to 75.2% in 2022 (Figure 1).

### 3.4. DTP, OPV, and PCV (Three Doses at 6, 10, and 14 Weeks)

In Cambodia, the proportion of children completing all three doses of DTP increased from 34.7% in 2000 to 76.0% in 2021, alongside a marked decline in unvaccinated prevalence from 39.7% to 10.0%. Similarly, the full coverage of OPV rose to 79.3% in 2021, and the unvaccinated prevalence declined to 7.0%. In 2021, 76.2% of children completed the PCV schedule, and only 8.9% had not received any PCV vaccines. In the Philippines, DTP and OPV coverage rose to 72.9% and 71.5% by 2013 but declined thereafter, to 69.3% and 67.4%, respectively, by 2022. Concurrently, the percentage of unvaccinated children for DTP increased from 8.8% to 15.8%, while the percentage of children who had not received any OPV vaccines rose from 10.6% to 16.5% between 2013 and 2022. In contrast, PCV coverage significantly increased from 36.3% in 2017 to 61.9% in 2022, though 21.1% of children remained unvaccinated (Figure 1).

### 3.5. Measles (Two Doses at 9 and 12/18 Months)

Completion of the two-dose measles vaccines remained lower in both countries, with 56.9% in Cambodia and 58.9% in the Philippines in 2022, despite a moderate increase from 44.8% in 2017 in the Philippines. In both countries, approximately one-fifth of children remained unvaccinated (Figure 1).

### 3.6. Disparity Between Urban and Rural Areas and by Wealth Groups

In Cambodia, urban–rural disparities in BCG, DTP, and OPV generally decreased between 2000 and 2021, with the smallest gaps observed in 2005, widening in 2010 before narrowing by 2021. There were moderate disparities for PCV (6.0%) and measles (5.7%), and a smaller gap for hepatitis B (birth dose) (2.0%) in 2021.

The Philippines showed a general decline in BCG, DTP, and OPV between 2003 and 2022. Gaps in hepatitis B (birth dose) and measles slightly narrowed from 11.5% to 9.9% and 5.1% to 2.6%, respectively, between 2017 and 2022. Notably, PCV coverage presented a reversal in disparity, with rural children having higher coverage than urban children (−5.4%) in 2017. However, by 2022, urban coverage surpassed rural coverage by 4.0% (Figure 2).

In Cambodia, wealth-related disparities in BCG, DTP, and OPV remained relatively stable from 2005 to 2014 but narrowed significantly from 2014 to 2021. By 2021, disparities in vaccines administered at birth (BCG and hepatitis B) were minimal. However, it was more pronounced for DTP and PCV, with disparities of 24.4pp (95% CI: 13.2–35.6) and 22.8pp (95% CI: 11.6–34.0), respectively. The largest disparity was observed for the measles vaccine, which is typically administered at older ages. Among children from the poorest group, only 47.7% (112/235) completed the vaccine on schedule, compared to 70.4% (88/125) from the richest group, revealing a wealth-related disparity of 25.2pp (95% CI: 0.4–50.1).

In the Philippines, wealth-related disparities in BCG, DTP, and OPV decreased between 2003 and 2017. From 2017 to 2022, however, disparities increased in all six vaccines, with the largest rise observed in PCV, from 7.5pp (95% CI: −5.8 to 20.8) to 38.7pp (95% CI: 26.0 to 51.3). By 2022, wealth-related disparities in DTP and measles vaccination reached 39.8pp (95% CI: 28.0–51.6) and 37.1pp (95% CI: 19.5–54.8), respectively. Significant wealth-related disparities persisted even for birth dose vaccines, with the difference of 32.1pp (95% CI: 22.1–42.0) for BCG and 42.9pp (95% CI: 32.7–53.2) for hepatitis B birth dose (Figure 3).

### 3.7. Determinants of Completing Vaccine Administration on Schedule

In Cambodia, after adjusting for all maternal demographic and socioeconomic characteristics, higher educational attainment and having only one or two children were associated with a higher likelihood of completing DTP, PCV, and measles vaccinations on schedule, compared to their counterparts. Older maternal age and living in wealthier households were also associated with a higher likelihood of completing DTP and PCV vaccinations. These factors similarly showed an increased likelihood of completing measles vaccination, but the differences were not statistically significant.

In the Philippines, higher maternal educational attainment, having fewer children, and belonging to wealthier households were associated with a higher likelihood of completing DTP, PCV, and measles vaccinations. Additionally, children of employed mothers had a 1.36-fold increased likelihood of completing DTP vaccination (95% CI: 1.12–1.65) and a 1.27-fold increased likelihood of completing PCV vaccination (95% CI: 1.06–1.52). However, no statistically significant association was observed between maternal age and the completion of any of these three vaccines (Table 2).

In both countries, logistic regression analyses for the other three types of vaccines (BCG, OPV, and hepatitis B birth dose) revealed similar results. The data are presented in Appendix A.

## 4. Discussion

Over the past two decades, BCG, DTP, and OPV completion rates have increased significantly in Cambodia but have slightly declined in the Philippines since 2017. In 2022, most Filipino children received the BCG and hepatitis B (birth dose) vaccines, and around two-thirds completed DTP, OPV, and PCV vaccinations according to the schedule, while this was lower than the rates in Cambodia. However, only half of the children completed the measles vaccination, and around one-fifth had not received any measles vaccines in both countries. In both Cambodia and the Philippines, urban–rural disparities in coverage for these six types of vaccinations decreased over time; however, inequalities among income groups persisted.

Cambodia has been a Gavi-eligible country since 2002 and adopted the WHO- and UNICEF-recommended Reaching Every District (RED) strategy in 2003 to empower communities and provide outreach services, addressing geographic accessibility barriers through performance-based incentives [17]. The combined efforts of the international community and domestic initiatives have successfully improved the coverage of several WHO-recommended childhood vaccines in Cambodia, a trend also observed in other low-income countries receiving Gavi support [18]. The smallest urban–rural disparity in vaccination coverage was observed in 2005. However, the gap widened again by 2010, potentially due to insufficient government financial support to sustain community-based initiatives and incentivize the performance of primary healthcare providers [19]. Cambodia transitioned to a lower-middle-income country in 2015, and the government has committed to mobilizing greater domestic resources to further enhance maternal and child health outcomes. In 2019, Cambodia launched the “First 1000 Days” initiative, offering cash support to poor women for antenatal, childbirth, postnatal, and child immunization services [20]. This may partially explain the narrowed urban–rural and wealth disparities in childhood vaccination coverage by 2021. With nearly all women giving birth in health facilities in Cambodia in 2021 [21], the disparity in coverage of birth vaccines, such as BCG and the hepatitis B birth dose, between urban and rural areas and across wealth groups was minimal. However, wealth-related disparities in the coverage of multiple-dose vaccines, including DTP, OPV, and PCV, as well as measles vaccines scheduled for older children, persisted, emphasizing the need to improve vaccine coverage among socioeconomically vulnerable groups to achieve equitable access to immunization.

Unlike Gavi-supported low-income countries, the Philippines’ NIP is mainly funded by the country government, supplemented by contributions from external partners [5]. In the Philippines, despite the 2011 mandate requiring vaccination of children under the NIP [22], the 2017 Dengvaxia controversy, which raised concerns about the vaccine associated with severe dengue in previously uninfected children, undermined public confidence in vaccines [23,24]. Additionally, the COVID-19 pandemic worsened immunization setbacks due to lockdowns and health service disruptions, driving a continued nationwide decline in coverage [25,26]. UNICEF data shows that between 2019 and 2021, the Philippines had the second-highest number of zero-dose children in East Asia and the Pacific, and the fifth globally [27]. This decline disproportionately affected socioeconomically vulnerable groups, as evidenced by a widening wealth-related disparity in vaccination coverage between 2017 and 2022. Specifically, PCV was initially introduced in selected regions with the aim of covering approximately 25–30% of eligible infants and was subsequently scaled up in 2020 with financial and technical support from Gavi’s MIC initiative [28,29]. However, the observed increase in PCV coverage from 2017 to 2022 revealed growing disparities between urban and rural areas, as well as pro-rich patterns, suggesting potential gaps in immunization services in less developed areas and among socioeconomically disadvantaged groups.

Despite evidence demonstrating the significant contribution of measles vaccination to child survival, complete measles vaccination coverage remains suboptimal in both Cambodia and the Philippines, particularly among socioeconomically vulnerable groups, a pattern similarly observed in other low- and middle-income countries [4]. Cambodia announced the elimination of measles in 2015; nevertheless, the country subsequently reported imported measles cases, which triggered a series of outbreaks [9]. Between 2018 and 2020, a large-scale measles outbreak predominantly affected children over six months of age, with the majority being either unvaccinated or having received only a single dose of the vaccine [30,31]. Similarly, the Philippines experienced one of the most severe measles outbreaks globally in 2019, largely attributed to low vaccination coverage driven by widespread mistrust in vaccines following the Dengvaxia controversy in the country [10]. Disparities in measles vaccine coverage also exist among different income groups, highlighting the need to place greater focus on socioeconomically vulnerable populations. Additionally, the previous studies reported that low measles vaccination rates in other low- and middle-income countries are linked to inadequate follow-up, negative experiences with health facilities, and limited knowledge of immunization schedules as well as vaccine hesitancy caused by misinformation [31,32,33].

Consistent with findings from other low- and middle-income countries [4,34], higher maternal educational attainment, having fewer children, and belonging to a wealthier household were consistently associated with timely and complete childhood vaccination in both study countries. These associations likely reflect greater maternal awareness, acceptance of immunization services, and caregiving capacity. Furthermore, maternal employment may empower women to take a more active role in securing better healthcare for their children (38). Therefore, the design of vaccination strategies should consider the social determinants of vaccination coverage to improve equity in childhood immunization.

Both Cambodia and the Philippines face significant challenges in ensuring sustainable and equitable vaccination coverage for children. Cambodia is currently entering Gavi’s transition phase, moving from reliance on external funding to mobilizing domestic resources for its NIP. During this transition, it is crucial to prevent backsliding in vaccine coverage, particularly among socially disadvantaged groups. Meanwhile, the Philippines is grappling with high and growing wealth inequities in vaccine coverage, highlighting critical gaps in the provision of immunization services. Hence, targeted pro-poor and community-based strategies, such as designing targeted education, reducing financial and logistical barriers to vaccination, empowering women, and implementing outreach efforts should be developed based on contextual analysis in both countries to ensure equitable access and that no one is left behind. Gavi’s continued support, along with the efforts of the international community, remains crucial to facilitate the design, implementation, and evaluation of strategies, as well as to ensure a smooth transition toward sustainable and equitable immunization targets.

### Strengths and Limitations

This study analyzed trends in childhood vaccination coverage over the past two decades in Cambodia, a Gavi-eligible country, and the Philippines, recently eligible for the Gavi MICs Approach, which provide insights into how policy frameworks impact vaccination coverage in low- and middle-income countries. However, several limitations must be acknowledged. Vaccination records may be affected by recall bias, as 16.1% Cambodia women and 27.8% of Filipino women reported their child’s vaccination history without a vaccination card. However, this analysis focused on the youngest children born within two years prior to the surveys, minimizing the likelihood of significant recall inaccuracies. In Cambodia, the relatively small number of children eligible to receive the second dose of the measles vaccine resulted in a wide 95% confidence interval when calculating wealth-related inequality (SII), introducing potential uncertainty into the results. In addition, the patterns of complete measles vaccination could not be observed during the first decade of the study periods in both study countries due to data unavailability. The overall low rate of complete measles vaccination in more recent years suggests the need for further exploration of the underlying reasons.

## 5. Conclusions

Both Cambodia and the Philippines have made progress in improving the coverage of several WHO-recommended childhood vaccines over time. However, significant inequalities remain across different economic groups. Furthermore, the low rate of complete measles vaccination raises serious concerns about the risks for measles outbreaks. The national immunization plan should strategically integrate international support with robust domestic commitment and efforts to address socioeconomic and geographic disparities, ensuring equitable vaccine access for all children.

## Figures and Tables

**Figure 1 vaccines-13-00907-f001:**
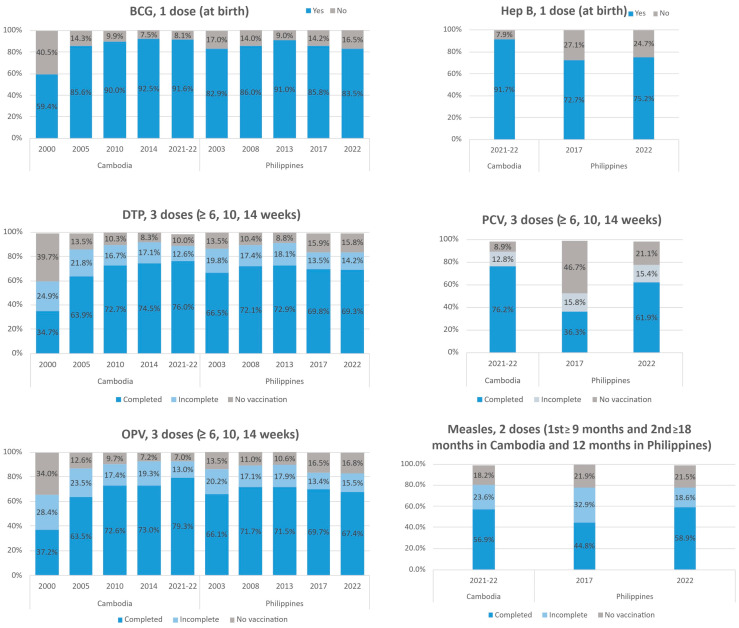
Vaccination coverage on schedule in Cambodia and the Philippines, 2000–2022.

**Figure 2 vaccines-13-00907-f002:**
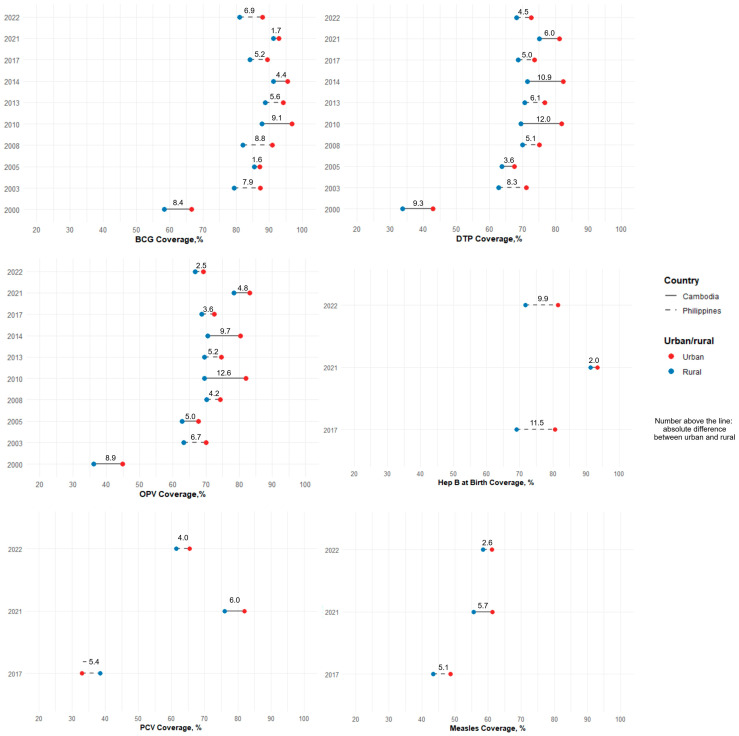
Vaccination coverage on schedule by urban and rural areas, 2000–2022.

**Figure 3 vaccines-13-00907-f003:**
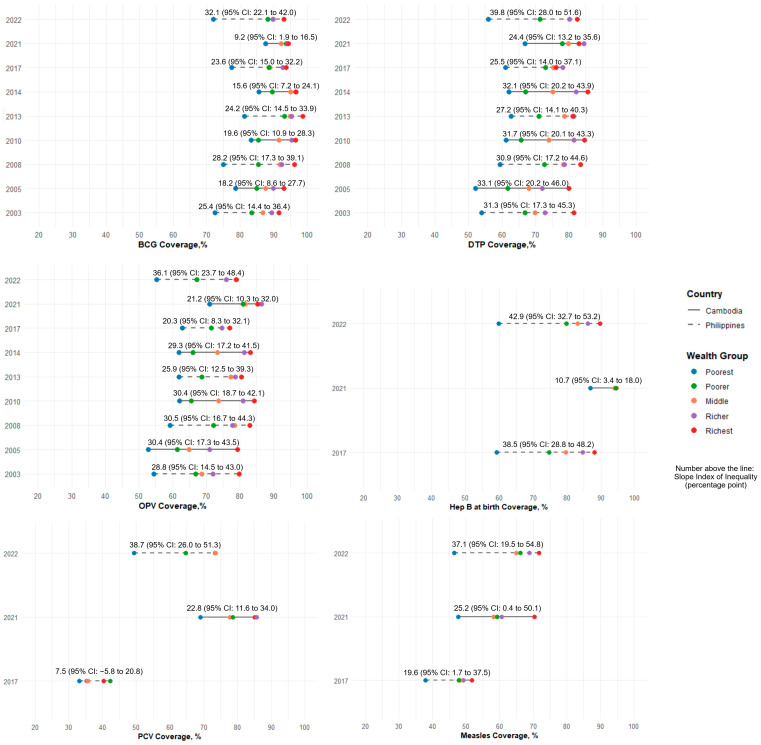
Vaccination coverage on schedule by wealth groups, 2000–2022.

**Table 1 vaccines-13-00907-t001:** Demographic and socioeconomic characteristics of women whose most recent child is aged 0–23 months in Cambodia, 2000–2021 (a) and the Philippines, 2003–2022 (b).

**(a) Cambodia**						
**Characteristics**	**2000** **(*n* = 2978)**	**2005** **(*n* = 3123)**	**2010** **(*n* = 3097)**	**2014** **(*n* = 2841)**	**2021–2022** **(*n* = 3348)**	***p*-Value**
Place of residence ^1^						<0.001
Urban	411 (13.8%)	643 (20.6%)	805 (26.0%)	772 (27.2%)	1099 (32.8%)	
Rural	2559 (85.9%)	2439 (78.1%)	2252 (72.7%)	2029 (71.4%)	2222 (66.4%)	
Maternal age						<0.001
<19	249 (8.4%)	245 (7.8%)	262 (8.5%)	268 (9.4%)	270 (8.1%)	
20–29	1379 (46.3%)	1723 (55.2%)	1969 (63.6%)	1762 (62.0%)	1783 (53.3%)	
30+	1350 (45.3%)	1155 (37.0%)	866 (28.0%)	811 (28.5%)	1295 (38.7%)	
Education						<0.001
No education	1089 (36.6%)	848 (27.2%)	608 (19.6%)	360 (12.7%)	401 (12.0%)	
Incomplete primary	1378 (46.3%)	1604 (51.4%)	1343 (43.4%)	1060 (37.3%)	994 (29.7%)	
Primary	493 (16.6%)	616 (19.7%)	988 (31.9%)	1147 (40.4%)	1502 (44.9%)	
Secondary and higher	18 (0.6%)	55 (1.8%)	158 (5.1%)	274 (9.6%)	451 (13.5%)	
Wealth index						<0.001
Poorest	-	873 (28.0%)	780 (25.2%)	659 (23.2%)	964 (28.8%)	
Poorer	-	715 (22.9%)	581 (18.8%)	528 (18.6%)	658 (19.7%)	
Middle	-	593 (19.0%)	520 (16.8%)	441 (15.5%)	611 (18.2%)	
Richer	-	475 (15.2%)	572 (18.5%)	506 (17.8%)	672 (20.1%)	
Richest	-	467 (15.0%)	644 (20.8%)	707 (24.9%)	443 (13.2%)	
Employment ^2^						<0.001
Did not work	636 (21.4%)	11 (0.4%)	747 (24.1%)	877 (30.9%)	1075 (32.1%)	
Worked	2335 (78.4%)	2347 (72.9%)	2347 (75.8%)	1953 (68.7%)	2159 (64.5%)	
Parity						<0.001
1	541 (18.2%)	861 (27.6%)	1104 (35.6%)	1163 (40.9%)	1099 (32.8%)	
2	541 (18.2%)	710 (22.7%)	837 (27.0%)	866 (30.5%)	1216 (36.3%)	
3+	1896 (63.7%)	1552 (49.7%)	1156 (37.3%)	812 (28.6%)	1033 (30.9%)	
Number of children eligible for vaccines on schedule						
One-dose vaccine, at birth	2978	3123	3097	2841	3348	
Three-dose vaccines, ≥14 weeks	2569	2742	2776	2518	2926	
Two-dose vaccines, ≥18 months	-	-	-	-	802	
**(b) The Philippines**						
**Characteristics**	**2003** **(*n* = 2592)**	**2008** **(*n* = 2443)**	**2013** **(*n* = 2718)**	**2017** **(*n* = 3809)**	**2022** **(*n* = 3000)**	***p*-Value**
Place of residence ^3^						<0.001
Urban	1190 (45.9%)	1016 (41.6%)	1101 (40.5%)	1218 (32.0%)	1059 (35.3%)	
Rural	1345 (51.9%)	1353 (55.4%)	1536 (56.5%)	2540 (66.7%)	1873 (62.4%)	
Maternal age						<0.001
<19	182 (7.0%)	221 (9.0%)	282 (10.4%)	365 (9.6%)	250 (8.3%)	
20 –29	1397 (53.9%)	1288 (52.7%)	1377 (50.7%)	2012 (52.8%)	1450 (48.3%)	
30 +	1013 (39.1%)	934 (38.2%)	1059 (39.0%)	1432 (37.6%)	1300 (43.3%)	
Education						<0.001
No education	57 (2.2%)	43 (1.8%)	47 (1.7%)	65 (1.7%)	41 (1.4%)	
Incomplete primary	355 (13.7%)	276 (11.3%)	289 (10.6%)	391 (10.3%)	189 (6.3%)	
Primary	834 (32.2%)	745 (30.5%)	748 (27.5%)	1010 (26.5%)	881 (29.4%)	
Secondary and higher	1346 (51.9%)	1379 (56.4%)	1634 (60.1%)	2343 (61.5%)	1889 (63.0%)	
Wealth index						<0.001
Poorest	745 (28.7%)	704 (28.8%)	800 (29.4%)	1339 (35.2%)	1074 (35.8%)	
Poorer	595 (23.0%)	589 (24.1%)	607 (22.3%)	903 (23.7%)	634 (21.1%)	
Middle	493 (19.0%)	444 (18.2%)	559 (20.6%)	669 (17.6%)	542 (18.1%)	
Richer	429 (16.6%)	417 (17.1%)	429 (15.8%)	531 (13.9%)	391 (13.0%)	
Richest	330 (12.7%)	289 (11.8%)	323 (11.9%)	367 (9.6%)	359 (12.0%)	
Employment ^4^						0.097
Did not work	1558 (60.1%)	1381 (56.5%)	1573 (57.9%)	2322 (61.0%)	1812 (60.4%)	
Worked	1025 (39.5%)	976 (40.0%)	1029 (37.9%)	1414 (37.1%)	1178 (39.3%)	
Parity						<0.001
1	720 (27.8%)	710 (29.1%)	821 (30.2%)	1120 (29.4%)	944 (31.5%)	
2	587 (22.6%)	536 (21.9%)	668 (24.6%)	978 (25.7%)	741 (24.7%)	
3+	1285 (49.6%)	1197 (49.0%)	1229 (45.2%)	1711 (44.9%)	1315 (43.8%)	
Number of children eligible for vaccines on schedule						
One-dose vaccine, at birth	2592	2443	2718	3809	3000	
Three-dose vaccines, ≥14 weeks	2297	2163	2415	3333	2677	
Two-dose vaccines, ≥12 months	-	-	-	1840	1473	

^1^ A total of 8 missing in 2000, 41 in 2005, 40 in 2010, 40 in 2014, and 27 in 2021; ^2^ 7 missing in 2000, 13 in 2005, 3 in 2010, 22 in 2014, and 79 in 2021; ^3^ 57 missing in 2003, 74 in 2008, 81 in 2013, 51 in 2017, and 68 in 2022; and ^4^ 9 missing in 2003, 86 in 2008, 116 in 2013, 73 in 2017, and 10 in 2022.

**Table 2 vaccines-13-00907-t002:** Factors associated with completing administration of DTP, PCV, and measles vaccine on schedule in Cambodia 2021 (a) and the Philippines (b).

**(a) Cambodia**						
	**DTP**		**PCV**		**Measles**	
**Variables**	**Adjusted ^1^ OR** **(95%CI)**	***p*-Value**	**Adjusted ^1^ OR** **(95%CI)**	***p*-Value**	**Adjusted ^1^ OR** **(95%CI)**	***p*-Value**
Maternal age						
<19	Ref.	Ref.	Ref.	Ref.	Ref.	Ref.
20–29	1.76 (1.25, 2.49)	0.001	1.73 (1.22, 2.46)	0.002	1.02 (0.55, 1.89)	0.9
30+	2.46 (1.65, 3.67)	<0.001	2.01 (1.34, 3.01)	<0.001	1.39 (0.69, 2.78)	0.4
Residence						
Rural	Ref.	Ref.	Ref.	Ref.	Ref.	Ref.
Urban	1.01 (0.80, 1.26)	1.0	1.02 (0.81, 1.28)	0.9	0.95 (0.67, 1.35)	0.8
Education						
No education	Ref.	Ref.	Ref.	Ref.	Ref.	Ref.
Incomplete primary	1.68 (1.27, 2.21)	<0.001	1.48 (1.12, 1.96)	0.006	1.99 (1.20, 3.32)	0.008
Primary	2.03 (1.52, 2.70)	<0.001	1.83 (1.37, 2.45)	<0.001	2.36 (1.43, 3.94)	<0.001
Secondary and higher	2.44 (1.61, 3.73)	<0.001	2.64 (1.71, 4.13)	<0.001	3.42 (1.75, 6.79)	<0.001
Wealth index						
Poorest	Ref.	Ref.	Ref.	Ref.	Ref.	Ref.
Poorer	1.46 (1.12, 1.91)	0.005	1.39 (1.07, 1.82)	0.02	1.29 (0.83, 2.00)	0.3
Middle	1.50 (1.13, 1.98)	0.005	1.20 (0.91, 1.58)	0.2	1.19 (0.74, 1.91)	0.5
Richer	1.98 (1.46, 2.70)	<0.001	1.95 (1.43, 2.68)	<0.001	1.31 (0.81, 2.11)	0.3
Richest	1.63 (1.13, 2.38)	0.01	1.64 (1.12, 2.42)	0.01	1.68 (0.95, 3.01)	0.08
Employment						
Did not work	Ref.	Ref.	Ref.	Ref.	Ref.	Ref.
Worked	1.02 (0.84, 1.23)	0.9	0.98 (0.80, 1.19)	0.8	0.80 (0.57, 1.12)	0.2
Parity						
3+	Ref.	Ref.	Ref.	Ref.	Ref.	Ref.
2	1.35 (1.06, 1.72)	0.01	1.31 (1.03, 1.67)	0.03	1.55 (1.04, 2.30)	0.03
1	1.67 (1.25, 2.24)	<0.001	1.55 (1.15, 2.08)	0.004	1.88 (1.18, 3.02)	0.008
**(b) The Philippines**						
	**DTP**		**PCV**		**Measles**	
**Variables**	**Adjusted ^1^ OR** **(95%CI)**	***p*-Value**	**Adjusted ^1^ OR** **(95%CI)**	***p*-Value**	**Adjusted ^1^ OR** **(95%CI)**	***p*-Value**
Maternal age						
<19	Ref.	Ref.	Ref.	Ref.	Ref.	Ref.
20–29	0.97 (0.67, 1.38)	0.9	0.95 (0.68, 1.33)	0.8	1.26 (0.79, 2.01)	0.3
30+	1.03 (0.69, 1.53)	0.9	1.05 (0.72, 1.53)	0.8	1.40 (0.84, 2.32)	0.2
Residence						
Rural	Ref.	Ref.	Ref.	Ref.	Ref.	Ref.
Urban	0.89 (0.73, 1.08)	0.2	0.90 (0.75, 1.08)	0.3	0.85 (0.67, 1.09)	0.2
Education						
No education	Ref.	Ref.	Ref.	Ref.	Ref.	Ref.
Incomplete primary	1.70 (0.80, 3.84)	0.2	1.22 (0.55, 2.95)	0.6	1.21 (0.47, 3.44)	0.7
Primary	3.12 (1.53, 6.79)	0.003	2.73 (1.29, 6.32)	0.01	2.33 (0.97, 6.22)	0.07
Secondary and higher	3.93 (1.92, 8.54)	<0.001	3.93 (1.85, 9.08)	<0.001	2.89 (1.20, 7.70)	0.02
Wealth index						
Poorest	Ref.	Ref.	Ref.	Ref.	Ref.	Ref.
Poorer	1.67 (1.32, 2.13)	<0.001	1.52 (1.20, 1.92)	<0.001	1.91 (1.40, 2.62)	<0.001
Middle	2.46 (1.86, 3.26)	<0.001	2.09 (1.61, 2.72)	<0.001	1.74 (1.24, 2.43)	0.001
Richer	2.36 (1.71, 3.31)	<0.001	1.99 (1.47, 2.72)	<0.001	2.05 (1.38, 3.08)	<0.001
Richest	2.59 (1.81, 3.74)	<0.001	1.82 (1.31, 2.52)	<0.001	2.24 (1.46, 3.46)	<0.001
Employment						
Did not work	Ref.	Ref.	Ref.	Ref.	Ref.	Ref.
Worked	1.36 (1.12, 1.65)	0.002	1.27 (1.06, 1.52)	0.01	1.13 (0.90, 1.43)	0.3
Parity						
3+	Ref.	Ref.	Ref.	Ref.	Ref.	Ref.
2	1.31 (1.03, 1.67)	0.03	1.26 (1.00, 1.58)	0.048	1.14 (0.85, 1.53)	0.4
1	1.24 (0.97, 1.61)	0.09	1.30 (1.02, 1.66)	0.03	1.46 (1.06, 2.02)	0.02

^1^ Adjusting for all explanatory variables.

## Data Availability

Data are available in a public, open access repository. All data used in this study are publicly available upon request at https://www.dhsprogram.com/data/Access-Instructions.cfm (accessed on 14 May 2024).

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
