# Peer review of "Bridging the Gap: Two Decades of Childhood Vaccination Coverage and Equity in Cambodia and the Philippines (2000–2022)"

_vaccines, 2025, doi:10.3390/vaccines13090907_

Round 1

Reviewer 1 Report

Comments and Suggestions for Authors

The authors work is a description of childhood immunization practices and the efficacy of deploying them in Cambodia and the Philippines. The authors look at factors such as the socioeconomic, maternal education and employment, size of family and urban vs rural setting. This is done by comparing multiple time periods when information was collected ranging from 2005 until 2021-2022. They have some limitations such as almost 20% of mother's interviewed needing to rely on memory to answer questions as which may lead to recall bias. An overall trend towards improvement in immunization coverage was detected until the most recent analysis. Reasons for this include the COVID-19 pandemic and the Dengvaxia controversy that occurred in the Philippines. This has led to a lower rate of vaccine series completion in the Philippines. Cambodia has solid birth coverage likely due to the fact that nearly all births occur in healthcare facilities.  Maternal education, improved family wealth as well as less children in a family have been associated with improved immunization administration. Generally, discrepancies exist between urban and rural settings with some retreat in the Philippines in the last reporting era. 

Overall, this is a well-written and presented paper. The authors analyze the data and do not try to make too many inferences regarding casuality but do present some possible investments for each of the countries to do to maintain progress and make improvements.

I do have a few suggestions:

  1. The authorship needs to be corrected with the "and" before the "PhD"
  2. Lines 145 and 146 discuss employment. You state one as employed percentage and another for unemployed. This is very clunky. Please describe both as % employed
  3. Figures 2 and 3 are nearly uninterpretable. They seem too dense. They do not fit on the page or the pdf or what I can see online. Additionally, the tables are very dense. I would recommend simplfying Figure 2 and 3. I believe that a legends are necessary for these figures to better help explain the data that is being represented. They have numerous colors and include data presented for the series as well as individual vaccines. This is the major change necessary to publish the paper. 

Author Response

The authors work is a description of childhood immunization practices and the efficacy of deploying them in Cambodia and the Philippines. The authors look at factors such as the socioeconomic, maternal education and employment, size of family and urban vs rural setting. This is done by comparing multiple time periods when information was collected ranging from 2005 until 2021-2022. They have some limitations such as almost 20% of mother's interviewed needing to rely on memory to answer questions as which may lead to recall bias. An overall trend towards improvement in immunization coverage was detected until the most recent analysis. Reasons for this include the COVID-19 pandemic and the Dengvaxia controversy that occurred in the Philippines. This has led to a lower rate of vaccine series completion in the Philippines. Cambodia has solid birth coverage likely due to the fact that nearly all births occur in healthcare facilities.  Maternal education, improved family wealth as well as less children in a family have been associated with improved immunization administration. Generally, discrepancies exist between urban and rural settings with some retreat in the Philippines in the last reporting era. 

Overall, this is a well-written and presented paper. The authors analyze the data and do not try to make too many inferences regarding casuality but do present some possible investments for each of the countries to do to maintain progress and make improvements.

I do have a few suggestions:

Comment 1: The authorship needs to be corrected with the "and" before the "PhD"

Response 1:Thank you for pointing this out. We have corrected with the “and” (line 4).

Comment 2:Lines 145 and 146 discuss employment. You state one as employed percentage and another for unemployed. This is very clunky. Please describe both as % employed

Response 2:Thank you for the suggestion. We have changed to both employed percentages (line 145–146).

Comment 3:Figures 2 and 3 are nearly uninterpretable. They seem too dense. They do not fit on the page or the pdf or what I can see online. Additionally, the tables are very dense. I would recommend simplfying Figure 2 and 3. I believe that a legends are necessary for these figures to better help explain the data that is being represented. They have numerous colors and include data presented for the series as well as individual vaccines. This is the major change necessary to publish the paper.

Response 3:Thank you for the suggestion. We have separated Table 1 by country to make it clearer (line 149–157) and adjusted the formatting of Figures 2 and 3 to conform to the journal template (line 196–197 and line 215–216). We have also extracted the data from Figures 2 and 3 and provided them in tables in the supplementary material.

Reviewer 2 Report

Comments and Suggestions for Authors

The paper written by Yanqin Zhang MPH and al. and entitled "Bridging the Gap: Two Decades of Childhood Vaccination Coverage and Equity in Cambodia and the Philippines (2000–2022)" is presented and easy to read and understand. The objectives, methodology and discussions are well presented. They are coherent to results obtained. Statistical investigations are well analyzed regarding the huge samples used in the study (almost 30000 samples).

Authors concluded important information regarding the vaccination coverage in theses two countries. They noted that Both Cambodia and the Philippines have made progress in improving the coverage of several WHO-recommended childhood vaccines over time. However, significant inequalities remain across different economic groups. Furthermore, the low rate of complete measles vaccination raises serious concerns about the risks for measles outbreaks. The national immunization plan should strategically integrate international support with robust domestic commitment and efforts to address socioeconomic and geographic disparities, ensuring equitable vaccine access for all children.

The article is suitable for publication after some minor revisions:

  • The scientific abreviations should be respected, for example Hepatitis B virus should be HBV and not Hep B....the same for the other viruses...
  • Figures 2 and 3 should be revised to be more complete and more clears. Some parts of theses Fig. are incompletes...

Author Response

The paper written by Yanqin Zhang MPH and al. and entitled "Bridging the Gap: Two Decades of Childhood Vaccination Coverage and Equity in Cambodia and the Philippines (2000–2022)" is presented and easy to read and understand. The objectives, methodology and discussions are well presented. They are coherent to results obtained. Statistical investigations are well analyzed regarding the huge samples used in the study (almost 30000 samples).

Authors concluded important information regarding the vaccination coverage in theses two countries. They noted that Both Cambodia and the Philippines have made progress in improving the coverage of several WHO-recommended childhood vaccines over time. However, significant inequalities remain across different economic groups. Furthermore, the low rate of complete measles vaccination raises serious concerns about the risks for measles outbreaks. The national immunization plan should strategically integrate international support with robust domestic commitment and efforts to address socioeconomic and geographic disparities, ensuring equitable vaccine access for all children.

The article is suitable for publication after some minor revisions:

Comment 1:The scientific abreviations should be respected, for example Hepatitis B virus should be HBV and not Hep B....the same for the other viruses...

Response 1: Thank you for pointing this out. We have verified that the vaccine abbreviations used in this manuscript are consistent with WHO documentation. Therefore, we did not make changes. (https://www.who.int/publications/m/item/table1-summary-of-who-position-papers-recommendations-for-routine-immunization)

Comment 2:Figures 2 and 3 should be revised to be more complete and more clears. Some parts of theses Fig. are incompletes...

Response 2:Thank you for pointing this out. We have adjusted the formatting of Figures 2 and 3 to conform to the journal template (line 196–197 and line 215–216). We have also extracted the data from Figures 2 and 3 and provided them in tables in the supplementary material.

Reviewer 3 Report

Comments and Suggestions for Authors

The investigators report an ERC-approved analysis of publicly-available Demographic and Health Survey data from two countries – Cambodia and the Philippines – to assess progress on vaccination coverage from 2000 to the most recent DHS data in each country and determine factors associated with progress. These countries represent lower-middle income countries, one of which is Gavi-eligible (Cambodia) and one of which is a Gavi “graduate.” Both countries implement all WHO-recommended vaccines except rotavirus vaccine, but DHS data included BCG, hepatitis B birth dose, DTP, OPV, PCV, and MCV. They found that coverage of these vaccines increased during the study period – continuously in Cambodia, and increase followed by decrease in the Philippines. They found significant urban-rural differences in coverage and significant coverage differences by several social determinants of health, for example, wealth and education attainment. They concluded that “Persistent socioeconomic disparities in childhood vaccination in low- and middle-income countries highlight the need for targeted pro-poor and community-based strategies to ensure equitable access.” They recommend that “the national immunization plan should strategically integrate international support with robust domestic commitment and efforts to address socioeconomic and geographic disparities, ensuring equitable vaccine access for all children.”

As the authors point out, EPI is a critically important program in all countries. Coverage is a key outcome for assessing program impact and strength. Financing is known to be an important factor for coverage. The authors selected two countries that are at the same income category (lower-middle income), but one is Gavi-eligible and about to transition, and the other is a Gavi graduate. This is a strategic choice of countries. Thus, the study and topic are clearly important. The data are publicly available and the surveys were conducted with similar methodologies. The analyses, while ecological, are appropriate to answer the study question; the analyses are well conducted. The limitations section is complete. The discussion section is thoughtful and well-articulated, and places the study findings into the international scientific literature. The conclusions are based on the data presented, as are the recommendations.

I have only minor suggestions to improve this very good manuscript.

Line 25 and six other places – A minor point. Timely vaccinations are usually termed “on schedule” rather than “in schedule.”

Line 44 – Reference is to an economic study on EPI impact, but the sentence is about the currently-recommended vaccines by WHO. A better reference is the WHO tables of recommended vaccines (https://www.who.int/publications/m/item/table1-summary-of-who-position-papers-recommendations-for-routine-immunization).

Paragraph starting on line 86 – It would be good to provide a reference to the actual data sets, as the authors do on lines 363-365.

Line 95 and following paragraph – It isn’t completely clear how coverage was calculated. Was a child considered vaccinated if he or she received a recommended vaccine by 24 months of age? Or was timely vaccination considered as vaccinated? If so, was a “grace period” allowed in the assessment of coverage for each vaccine? For example, if a child was one week late in receiving a recommended dose, was he or she counted as unvaccinated? The methods section should specify more precisely the coverage measures so that another researcher can reproduce the study.

Line 110 – “explainable” should be “explanatory.”

Lines 145-146 – The sentence, “64.5% of Cambodian women employed in 2021, while 60.4% of Filipino women were unemployed in 2022” compares the % employed in one country and the % unemployed in the other country. It would be better to compare on the same variable.

Figures 2 and 3 show incompletely. The figures do not fit completely in the pages and the color coding cannot be seen, rendering the figures difficult to understand.

Line 291 – It would be worth mentioning that Cambodia lost measles elimination status due to these outbreaks (Vaccines 2024, 12(7), 821; https://doi.org/10.3390/vaccines12070821).

Line 388 – The DHS referenced in reference 11 prefers to be references as “Philippine Statistics Authority (PSA) and ICF. 2023. 2022 Philippine National Demographic and Health Survey (NDHS): Final Report. Quezon City, Philippines, and Rockville, Maryland, USA: PSA and ICF.” The authors should clarify this reference (and reference 10).

Author Response

The investigators report an ERC-approved analysis of publicly-available Demographic and Health Survey data from two countries – Cambodia and the Philippines – to assess progress on vaccination coverage from 2000 to the most recent DHS data in each country and determine factors associated with progress. These countries represent lower-middle income countries, one of which is Gavi-eligible (Cambodia) and one of which is a Gavi “graduate.” Both countries implement all WHO-recommended vaccines except rotavirus vaccine, but DHS data included BCG, hepatitis B birth dose, DTP, OPV, PCV, and MCV. They found that coverage of these vaccines increased during the study period – continuously in Cambodia, and increase followed by decrease in the Philippines. They found significant urban-rural differences in coverage and significant coverage differences by several social determinants of health, for example, wealth and education attainment. They concluded that “Persistent socioeconomic disparities in childhood vaccination in low- and middle-income countries highlight the need for targeted pro-poor and community-based strategies to ensure equitable access.” They recommend that “the national immunization plan should strategically integrate international support with robust domestic commitment and efforts to address socioeconomic and geographic disparities, ensuring equitable vaccine access for all children.”

As the authors point out, EPI is a critically important program in all countries. Coverage is a key outcome for assessing program impact and strength. Financing is known to be an important factor for coverage. The authors selected two countries that are at the same income category (lower-middle income), but one is Gavi-eligible and about to transition, and the other is a Gavi graduate. This is a strategic choice of countries. Thus, the study and topic are clearly important. The data are publicly available and the surveys were conducted with similar methodologies. The analyses, while ecological, are appropriate to answer the study question; the analyses are well conducted. The limitations section is complete. The discussion section is thoughtful and well-articulated, and places the study findings into the international scientific literature. The conclusions are based on the data presented, as are the recommendations.

I have only minor suggestions to improve this very good manuscript.

Comment 1:Line 25 and six other places – A minor point. Timely vaccinations are usually termed “on schedule” rather than “in schedule.”

Response 1:Thank you for pointing this out. We have revised to ‘on schedule’ (line 25, 96, 159, 168, 197, 204, 216, 217, 220, 235).

Comment 2:Line 44 – Reference is to an economic study on EPI impact, but the sentence is about the currently-recommended vaccines by WHO. A better reference is the WHO tables of recommended vaccines (https://www.who.int/publications/m/item/table1-summary-of-who-position-papers-recommendations-for-routine-immunization).

Response 2:Thank you for the suggestion. We have changed the reference (line 44).

Comment 3:Paragraph starting on line 86 – It would be good to provide a reference to the actual data sets, as the authors do on lines 363-365.

Response 3:Thank you for the suggestion. We have added the reference (Line 88).

Comment 4:Line 95 and following paragraph – It isn’t completely clear how coverage was calculated. Was a child considered vaccinated if he or she received a recommended vaccine by 24 months of age? Or was timely vaccination considered as vaccinated? If so, was a “grace period” allowed in the assessment of coverage for each vaccine? For example, if a child was one week late in receiving a recommended dose, was he or she counted as unvaccinated? The methods section should specify more precisely the coverage measures so that another researcher can reproduce the study.

Response 4:Thank you for pointing this out. In our analysis, we considered a child vaccinated if they had reached the eligible age for a specific vaccine (e.g., over 14 weeks for DTP) and had received the vaccine by the time of the survey. We have revised the content in the method part (line 107–108).

Comment 5:Line 110 – “explainable” should be “explanatory.”

Response 5:Thank you for pointing this out. We have revised it (line 110).

Comment 6:Lines 145-146 – The sentence, “64.5% of Cambodian women employed in 2021, while 60.4% of Filipino women were unemployed in 2022” compares the % employed in one country and the % unemployed in the other country. It would be better to compare on the same variable.

Response 6:Thank you for the suggestion. We have changed to report both employed percentage (line 145–146).

Comment 7:Figures 2 and 3 show incompletely. The figures do not fit completely in the pages and the color coding cannot be seen, rendering the figures difficult to understand.

Response 7:Thank you for pointing this out. We have adjusted the formatting of Figures 2 and 3 to conform to the journal template (line 196–197 and line 215–216). We have also extracted the data from Figures 2 and 3 and provided them in tables in the supplementary material.

Comment 8:Line 291 – It would be worth mentioning that Cambodia lost measles elimination status due to these outbreaks (Vaccines 2024, 12(7), 821; https://doi.org/10.3390/vaccines12070821).

Response 8:Thank you for the suggestion. We have added this reference (line 296).

Comment 9:Line 388 – The DHS referenced in reference 11 prefers to be references as “Philippine Statistics Authority (PSA) and ICF. 2023. 2022 Philippine National Demographic and Health Survey (NDHS): Final Report. Quezon City, Philippines, and Rockville, Maryland, USA: PSA and ICF.” The authors should clarify this reference (and reference 10).

Response 9:Thank you for pointing this out. We have revised the DHS final report references for both the Philippines and Cambodia (line 399–402).

Reviewer 4 Report

Comments and Suggestions for Authors

This study assessed coverage of WHO-recommended childhood vaccines in Cambodia and the Philippines, focusing on urban-rural and wealth disparities, and examined maternal demographic and socio-economic factors influencing vaccination coverage.

Below are my comments

  1. While this article is interesting and timely, the authors should elaborate more on the issue of disparities and equity observed. This can serve as a yardstick for policymaking and interventions. This can be added to the discussion or a whole section dedicated to this subject and decision-making.
  2. Change your citation style to close bracket—[].
  3. Tables 1 and 2 should be moved to the appendix. This will help readability and flow of the work. 

Author Response

This study assessed coverage of WHO-recommended childhood vaccines in Cambodia and the Philippines, focusing on urban-rural and wealth disparities, and examined maternal demographic and socio-economic factors influencing vaccination coverage.

Below are my comments

Comment 1: While this article is interesting and timely, the authors should elaborate more on the issue of disparities and equity observed. This can serve as a yardstick for policymaking and interventions. This can be added to the discussion or a whole section dedicated to this subject and decision-making.

Response 1: Thank you for the suggestion. We have revised the discussion and made minor structural adjustments to more clearly elaborate on the issues of equity (line 271–272, 280–282 and 302–303).

Comment 2: Change your citation style to close bracket—[].

Response 2: Thank you for pointing this out. We have revised to close bracket.

Comment 3: Tables 1 and 2 should be moved to the appendix. This will help readability and flow of the work. 

Response 3: Thank you for the suggestion. Tables 1 and 2 are considered core tables, so we have retained them in the main manuscript. However, we have revised their formatting to improve clarity (line 151–157 and 237–240). If the journal prefers to place them in the Appendix for layout reasons, we are also open to that.